# Angiogenesis Driven by the CEBPD–hsa-miR-429–VEGFA Signaling Axis Promotes Urothelial Carcinoma Progression

**DOI:** 10.3390/cells11040638

**Published:** 2022-02-11

**Authors:** Ti-Chun Chan, Chung-Hsi Hsing, Yow-Ling Shiue, Steven K. Huang, Kun-Lin Hsieh, Yu-Hsuan Kuo, Chien-Feng Li

**Affiliations:** 1Chi Mei Medical Center, Tainan 71004, Taiwan; 090807@nhri.edu.tw (T.-C.C.); hsing@mail.chimei.org.tw (C.-H.H.); cmh7530@mail.chimei.org.tw (S.K.H.); 990707@mail.chimei.org.tw (K.-L.H.); a10901@mail.chimei.org.tw (Y.-H.K.); 2National Health Research Institutes, National Institute of Cancer Research, Tainan 70456, Taiwan; 3Institute of Precision Medicine, National Sun Yat-sen University, Kaohsiung 804201, Taiwan; shirley@imst.nsysu.edu.tw; 4Department of Biotechnology, Chia Nan University of Pharmacy and Science, Tainan 71710, Taiwan; 5College of Pharmacy and Science, Chia Nan University, Tainan 71710, Taiwan

**Keywords:** CEBPD, hsa-miR-429, VEGFA, urothelial carcinoma, angiogenesis

## Abstract

Background and Purpose: This research aimed to excavate the alternative mechanism of CEBPD on tumor growth and explore the biological significance of the CEBPD/hsa-miR-429/VEGFA axis on angiogenesis in urothelial carcinoma (UC). Methods: Quantitative RT-PCR, immunoblotting assay and tube formation examined the effect of hsa-miR-429 mimic or/and inhibitor on VEGFA expression and angiogenesis in CEBPD-overexpressing UC-derived cells. The association between CEBPD, hsa-miR-429, VEGFA and microvascular density (MVD) and clinical outcome were evaluated in 296 patients with UBUC and 340 patients with UTUC, respectively. Results: The increase in the transcript and protein of VEGFA as well as HUVECs tube formation was diminished upon the treatment of hsa-miR-429 mimic in CEBPD-overexpressing BFTC909 and TCCSUP. Nevertheless, the inhibited regulation of hsa-miR-429 mimic on the expression of VEGFA and ability of HUVECs tube formation was rescued by the combined incubation with hsa-miR-429 inhibitor in these two UC-derived cell lines. Furthermore, the clinical correlations showed that the higher level of VEGFA or MVD has a positive correlation with the expression of CEBPD and a negative relation to hsa-miR-429 and leads to tumor aggressiveness with worse disease-specific, metastasis-free survival in UBUC and UTUC cohorts. Conclusions: We decipher the oncogenic mechanism of CEBPD on angiogenesis through the hsa-miR-429 inhibition to stabilize the expression of VEGFA in UC. The novel research unveiled the modulation of the CEBPD/hsa-miR-429/VEGFA axis on the progression of UC and could be accessible to theranostic biomarkers.

## 1. Introduction

More than 90–95% of urothelial carcinoma (UC, also known as transitional cell carcinoma) emerge in bladder urothelial carcinoma (UBUC), and less than 10% of UC originate in upper tract urothelial carcinoma (UTUC, renal pelvis and ureter) in the United States [1]. Nevertheless, a higher prevalence of UTUC exists in Taiwan than in Western countries on account of the widespread use of aristolochic acid (AA) in traditional Chinese herbology before it was identified as a carcinogen for the development of kidney disease and UTUC [2,3]. UBUC was ranked as the tenth most commonly diagnosed malignancy, with 573,000 new cases and 213,000 deaths worldwide according to the estimation of the World Health Organization (WHO) [4]. For classification, UBUC can be categorized into papillary urothelial neoplasm of low malignant potential (PUNLMP), low-grade papillary UC and high-grade papillary UC based on cellular morphology in the grading system [5]. In the American Joint Committee on Cancer (AJCC) TNM (tumor, node, metastasis) staging system, UBUC is defined as non-muscle-invasive bladder cancer (NMIBC) including carcinoma in situ (CIS or Tis), Ta, T1 and muscle-invasive bladder cancer (MIBC) comprising T2, T3, T4, due to its capacity of muscular aggressiveness [6].

NMIBC accounts for approximately 75% of UBUC and is characterized by a favorable survival rate but high recurrence and potential MIBC progression [7]. The standard management of NMIBC is transurethral resection, intravesical chemotherapy and immunotherapy with Bacillus Calmette–Guérin (BCG) [8]. MIBC is more advanced and requires multimodality treatment combined with chemotherapy, radical cystectomy and radical radiotherapy [9]. Notwithstanding aggressive management, over 50% of patients with MIBC are prone to proceed to metastasis with a survival time of 13 to 20 months [7]. Accordingly, a widespread understanding of the molecular signaling mechanism regulating UC to develop innovative biomarkers is imperative and beneficial to optimize cure options and avoid inferior clinical outcomes.

In our published research, we showed that the regulation of CEBPD (CCAAT/enhancer-binding protein delta) on tumor growth is partially through the transcriptional suppression of hsa-miR-429 to promote glycolysis-related HK2 expression in UTUC- and UBUC-derived cell lines (BFTC909, TCCSUP) [10]. CEBPD, a transcription factor that belongs to the CCAAT/enhancer-binding protein family, is not only involved in various normal biological processes including cell proliferation, differentiation, metabolism, immune responses, growth arrest and cell death [11,12] but also displays as a dual functional modulator to diminish or provoke cancer development [13,14,15]. hsa-miR-429, a member of the miR-200 family, functions as a tumor suppressor or oncomiR, depending on different types of cancer [16]. Our previous study exhibited that hsa-miR-429 is greatly expressed in the clinical specimens with aggressive UBUC and along with low survival rate in patients with UTUC and UBUC. Therefore, it is worthwhile to deeply explore alternative mediation of the CEBPD–hsa-miR-429 axis to shed light on underlying molecular mechanisms of UC progression.

Several genes that refer to carcinogenesis have been verified as the targets for hsa-miR-429 (Wang et al., 2019). VEGFA could be a possible target for hsa-miR-429 through the computational algorithm prediction ([10], Appendix A). VEGFA belongs to the VEGF family (vascular endothelial growth factor; VEGFA, VEGFB, VEGFC, VEGFD, placenta growth factor [PGF]) and represents the best characteristics for angiogenesis to promote tumor growth and systemic metastasis (Carmeliet and Jain, 2011; Peach et al., 2018). Recently, CEBPD has been manifested to attribute chemotherapy-triggered angiogenesis to VEGF correlation and confer metastasis of lung cancer (Riedel et al., 2004; Chi et al., 2021), but the underlying mechanism is not yet well addressed. Accordingly, the study aimed to decipher the crosstalk of the CEBPD–hsa-miR-429–VEGFA axis on angiogenesis related to the progress of bladder cancer.

## 2. Materials and Methods

### 2.1. Cell Culture

Human UTUC- and UBUC-derived cell lines, namely BFTC909 and TCCSUP, respectively, were used in this study. BFTC909 was purchased from Food Industry Research and Development (FIRDI, Zhongzheng, Taipei, Taiwan), while TCCSUP was bought from American Type Tissue Culture Collection (American Type Culture Collection [ATCC], Manassas, VA, USA). Both BFTC909 and TCCSUP were maintained in the DMEM (Dulbecco’s Modified Eagle Medium) with 10% FBS (fetal bovine serum), 1% P/S (Penicillin–Streptomycin) and was incubated in a 37 °C humidified incubator with 5% CO_2_. Medium, serum and antibiotics were ordered from Gibco (Waltham, MA, USA).

### 2.2. Exogenous CEBPD Overexpression in BFTC909 and TCCSUP

Phoenix-AMPHO cell line (American Type Culture Collection [ATCC], Manassas, VA, USA) was used for the production of lentiviral particles containing the *CEBPD* gene. Briefly, a mixture including viral vector for *CEBPD*, *psPAX2* (lentiviral packaging plasmid), *pMD2.G* (lentiviral envelope plasmid) purchased from National Core Facility for Biopharmaceuticals (NCFB, Nangang, Taipei, Taiwan) and PolyJet™ transfection reagents (SignaGen^®^ Laboratories, Frederick, MD, USA) diluted in the Opti-MEM^®^ I Reduced-Serum Medium (Gibco, Waltham, MA, USA) was delivered into Phoenix-AMPHO cells. After 48 h post-transfection, the viral supernatant was harvested and infected BFTC909 and TCCSUP. The UC cell lines with CEBPD overexpression were selected as stable clones after the incubation of the growth medium with 2 µg/mL puromycin (Gibco, Waltham, MA, USA).

### 2.3. Transfection of hsa-miR-429 Mimic and Inhibitor

miRIDIAN microRNA Mimic Negative Control, miRIDIAN microRNA Human hsa-miR-429 mimic and miRIDIAN microRNA Human hsa-miR-429 inhibitor were all purchased from Horizon Discovery Group plc (Waterbeach, Cambridge, UK). miRIDIAN microRNA Human hsa-miR-429 inhibitor is a synthetically single-strand inhibitor made up of chemically enhanced RNA oligonucleotides designed to bind and to sequester the complimentary, mature hsa-miR-429 strand. miRNAs were transfected into cells by using jetPRIME^®^ in vitro DNA and siRNA transfection reagent (Polyplus-transfection SA, Illkirch-Graffenstaden, France). Concisely, 50 nM of miRNA Mimic Negative Control or has-miR-429 mimic or inhibitor was diluted in the jetPRIME^®^ buffer and was mixed with jetPRIME^®^ reagent for 10 min at room temperature. After that, the transfection mix mentioned above was added to the cells with a fresh medium. Further experiments were executed after 72 h post-transfection.

### 2.4. Total RNA Isolation and Evaluation by Real-Time Quantitative RT-PCR

Total RNA was isolated from BFTC909 and TCCSUP by using Quick-RNA™ Miniprep Kit (Zymo Research, Irvine, CA, USA) according to the manufacturer’s instructions. A proper amount of RNA was applied for cDNA synthesis by utilizing the Maxima First Strand cDNA Synthesis Kit referring to its manual (Gibco, Waltham, MA, USA) in a thermal cycler (Takara, Shiga, Japan). cDNA was mixed with predesigned TaqMan assay probe (VEGFA (Hs00900055_mL), Applied Biosystems; POLR2A, TopGen, Kaohsiung, Taiwan), using TaqMan Fast Advanced Master Mix (Thermo Scientific, Waltham, MA, USA) and was subjected to a StepOne Plus System machine to conduct quantitative RT-PCR for the mRNA level evaluation under 2^−ΔΔCT^ formula. The data were shown as the fold change in *VEGFA* gene expression in the experimental group normalized to the *POLR2A* and relative to the control group.

### 2.5. Immunoblotting Assay

Total protein was isolated from BFTC909 and TCCSUP by using PRO-PREP Protein Extraction Solution (iNtRON Biotechnology, Seongnam, Korea) according to its manual. A proper amount of proteins was separated dependent on their molecular weight in a Bis-Tris Precast Gel (Merck Millipore, Burlington, MA, USA) and was transferred onto an Immobilon-P PVDF membrane (Merck Millipore, Burlington, MA, USA). The PVDF membrane (Merck Millipore, Burlington, MA, USA) was blocked with skim milk (MilliporeSigma, Burlington, MA, USA) followed by the incubation of anti-VEGFA primary antibody (ab214424, Abcam, Cambridge, UK) or anti-GAPDH primary antibody (ab181602, Abcam, Cambridge, UK) diluted with 5% milk in TBST. After washing, the membrane was incubated with 5% milk in TBST containing HRP Donkey anti-rabbit IgG (BioLegend^®^, San Diego, CA, USA). Afterward, the VEGFA or GAPDH protein was visualized by using enhanced chemiluminescence (ECL; Thermo Scientific).

### 2.6. Luciferase Reporter Assays

pMIR-REPORT-VEGFA-WT-3′UTR or pMIR-REPORT-VEGFA-mir-429-3p-mutant -3′UTR (Firefly luciferase reporter, TopGen) was co-transfected with pGL4.74[hRluc/TK] (Renilla luciferase reporter as internal control, Promega, Madison, WI, USA) into cells. After 72 h of incubation, the relative light units were detected by using Dual-Glo^®^ Luciferase Assay System (Promega) with an ELISA reader (Promega, Madison, WI, USA). Firefly luciferase luminescence was normalized to Renilla luciferase luminescence to express the activity of WT or mutant VEGFA-3′UTR reporter.

### 2.7. HUVECs Tube Formation Assay

HUVEC (Human umbilical vein endothelial cell) was used for the tube formation assay. Concisely, Matrigel^®^ Basement Membrane Matrix (Corning, Corning, NY, USA) was loaded into each inner well of the µ-Slide Angiogenesis (Ibidi, Fitchburg, WI, USA) and put in the 37 °C cell culture incubator for 30 min. Then 50 µL cell suspension containing 7 × 10^3^ cells in 25 µL endothelial cell medium (ECM) with 2% FBS and 25 µL conditioned medium was seeded into each well of µ-Slide. After 5 h of incubation, the image of tube formation was photographed by phase-contrast microscopy.

### 2.8. Human Tumor Tissue Specimens

295 UBUC and 340 UTUC specimens from the biobank of Chi Mei Medical Center were used for the immunohistochemistry (IHC) and In Situ Hybridization Detection of miRNA. The archived tumor specimens were obtained from surgery with curative intent between January 1996 and May 2004 as our previous study described [17]. This research was approved by the institutional review board of Chi Mei Medical Center (IRB10207-001).

### 2.9. Immunohistochemistry (IHC)

Paraffin-embedded (FFPE) blocks of the clinical sample were microdissected (4 µm) and put onto precoated slides. Wax melting of the slides was conducted in a 65 °C oven for 20 min. After that, slides were deparaffinized with xylene and rehydrated with ethanol from high concentration to low concentration (100%, 95%, 75%) and distilled water, sequentially. The antigen retrieval process was treated with citrate buffer (10 mM citric acid, pH 6) in a microwave oven for 20 min. After cooling, endogenous peroxidase of slides was eliminated with Peroxidase-Blocking Solution (Dako, Glostrup, Hovedstaden, Denmark), followed by the incubation of the indicated primary antibody (CEBPD, NB110-85519; VEGFA, ab214424; PECAM1, ab28364) for one hour at room temperature. After washing with PBS, slides were incubated with peroxidase-conjugated secondary antibody reagent (REAL™ EnVision™/HRP, Rabbit/Mouse (ENV); Dako, Glostrup, Hovedstaden, Denmark) for 30 min at room temperature. The targeted protein was visualized with EAL™ DAB+ Chromogen diluted in REAL™ Substrate Buffer (Dako, Glostrup, Hovedstaden, Denmark). The nuclear location was detected with Gill’s hematoxylin. Afterward, slides were dehydrated with ethanol (75%, 95% and 100%) and mounted. The image was captured by phase-contrast microscopy and examined by an expert pathologist (CF Li) in Chi Mei Medical Center (Tainan, Taiwan). Computerized analysis was performed to quantify tumoral microvessel density (MVD) of the PECAM1-labeled vessels by ImageJ software, which calculated the percentage of vessel areas in each representative field.

### 2.10. In Situ Hybridization Detection of miRNA Using LNA™ Oligonucleotides

Hsa-miR429 was detected by the IsHyb In Situ Hybridization Kit (BioChain, Newark, CA, USA) according to its manufacturer’s instructions. Concisely, after wax melting and deparaffinization as the IHC procedure mentioned above, slices were treated with 4% DEPC–paraformaldehyde followed by a wash with DEPC–PBS. Then samples were treated with Proteinase K and were washed with DEPC–PBS. Following the treatment with prehybridization solution, the slices were incubated with a digoxigenin-labeled LNA probe against hsa-miR-429 overnight. The slides were washed by 2X SSC, 1.5X SSC and 0.2X SSC once serially and were incubated with AP-conjugated anti-digoxigenin antibody. After that, the samples were incubated with a mixture of nitro-blue tetrazolium (NBT) and 5-bromo-4-chloro-3′-indolyphosphate (BCIP) diluted in 1X alkaline phosphatase buffer in the dark. The Nuclear Fast Red solution (MilliporeSigma, Burlington, MA, USA) was applied for nuclear counterstaining. After the dehydration procedure with ethanol from low concentration to high concentration (75%, 95% and 100%), the slices were mounted and examined.

### 2.11. Statistics

Chi-square test, Spearman’s rank correlation coefficient and the Mann–Whitney U test using SPSS software (version 14.0, IBM, Armonk, NY, USA) were utilized to estimate the associations and correlations between CEBPD, VEGFA, hsa-miR-429, MVD, clinicopathologic variables and the comparisons for several functional studies. The expression status of targeted proteins and hsa-miR-429 were evaluated in line with our previous works [10]. Disease-specific survival (DSS) and metastasis-free survival (MeFS) were estimated and plotted through log-rank tests and Kaplan–Meier curves, respectively. Independent prognostic impacts of selected parameters were estimated by a multivariate Cox proportional hazards model. Differences with a two-tailed *p*-value lower than 0.05 were considered significant for all analyses.

## 3. Results

### 3.1. hsa-miR-429 Inhibits the Expression of VEGFA and Angiogenesis in BFTC909 and TCCSUP by Directly Targeting VEGFA mRNA

Because downregulated hsa-miR-429 in CEBPD-overexpressing cell lines has been unveiled in our previous study [10], we next examine the correlation between CEBPD, hsa-miR-429 and VEGFA in UC cell lines. Figure 1 and Figure 2 showed that the mRNA and protein level of VEGFA were promoted after exogenous CEBPD overexpression in BFTC909 and TCCSUP compared to those of the mock group. The treatment of hsa-miR-429 mimic diminished the capacity of CEBPD on the upregulation of VEGFA mRNA and protein expression. While the repressed mRNA and protein of VEGFA aroused from the hsa-miR-429 mimic in CEBPD-overexpressing cell lines were retrieved by the has-miR-429 inhibitor.

To confirm if the *VEGFA* transcript is directly targeted by the hsa-miR-429, BFTC909 (Figure 3A) and TCCSUP (Figure 3B) cell lines were co-transfected with WT (wild type)- or mutant-*VEGFA* 3′-UTR firefly luciferase reporter and miRNA Mimic Negative Control or hsa-miR-429 mimic or both hsa-miR-429 mimic and inhibitor. The hsa-miR-429 mimic markedly inhibited the luciferase activity of the WT-*VEGFA* 3′-UTR compared to BFTC909 and TCCSUP with miRNA Mimic Negative Control. The decreasing luciferase activity of the WT-*VEGFA* 3′-UTR caused by hsa-miR-429 mimic was rescued after treating with the combination of hsa-miR-429 mimic and inhibitor in these two cell lines. However, the transfection of miRNA Mimic Negative Control, hsa-miR-429 mimic, the combination of hsa-miR-429 mimic and inhibitor all failed to affect the firefly luciferase activity once BFTC909 and TCCSUP were transfected with mutant-*VEGFA* 3′-UTR reporter. The above data assumed that hsa-miR-429 directly targets the *VEGFA* transcript to inhibit its mRNA and protein expression.

Given that VEGFA is a critical factor for the angiogenesis in cancers, the ability of HUVECs tube formation was determined in the mock-CEBPD-overexpressing BFTC909 and TCCSUP cells treated with miRNA mimic negative control or hsa-miR-429 mimic or/and hsa-miR-429 inhibitor. Figure 4 shows that conditioned medium from the CEBPD-overexpressing BFTC909 and TCCSUP notably enhanced HUVECs tube formation compared to those from the mock group. Nevertheless, the ability of HUVEC-based tube formation was hampered after the treatment of conditioned medium from the CEBPD-overexpressing BFTC909 and TCCSUP transfected with hsa-miR-429 mimic than those from the UC-derived cells with CEBPD overexpression, while it was rescued upon treating with medium collected from transfecting hsa-miR-429 mimic and inhibitor simultaneously to ectopic CEBPD overexpression cells.

### 3.2. UBUC and UTUC Patients with High Expression of VEGFA and High Microvascular Density Are Usually Accompanied by Higher CEBPD Expression, Lower hsa-miR-429 and Inferior Survival Outcomes

Aside from in vitro research, we then investigated the correlations between CEBPD, hsa-miR-429 and VEGFA expression and their associations with microvascular density and clinical aggressiveness in UBUC and UTUC cohorts comprised of cancer tissue samples from 295 and 340 patients, respectively. Of note, CEBPD high expression positively correlated with higher VEGFA expression and more PECAM1-labeled MVD and negatively correlated with hsa-miR-429 in both UBUC and UTUC (Figure 5). Moreover, it turned out that higher expression of CEBPD, VEGFA and PECAM1-labeled MVD but lower levels of hsa-miR-429 were associated with numerous adverse clinicopathological factors. UBUCs and UTUCs with high expression of VEGFA usually have a positive correlation with high pathological staging (T, *p* < 0.001 and *p <* 0.001), high nodal metastasis (*p* < 0.001 and *p* < 0.001), high histological grade (*p* < 0.001 and *p* = 0.015), the presence of vascular invasion (*p* < 0.001 and *p* < 0.001) and perineural invasion (*p* < 0.001 and *p* < 0.001), high mitotic rate (*p* < 0.001 and *p* = 0.022) and high CEBPD expression (*p* < 0.001 and *p* < 0.001) but less hsa-miR-429 expression (*p* < 0.001 and *p* < 0.001) (Table 1 and Table 2). Similarly, high microvascular density is also accompanied by high pathological staging (T, *p* < 0.001 and *p* < 0.001, high nodal metastasis (*p* < 0.001 and *p* < 0.001), high histological grade (*p* < 0.001 and *p* < 0.001), the presence of vascular invasion (*p* < 0.001 and *p* = 0.001), perineural invasion (*p* < 0.001 and *p* < 0.001), high CEBPD expression (*p* < 0.001 and *p* < 0.001) (Figure 6A,D), high VEGFA expression (*p* < 0.001 and *p* < 0.001) (Figure 6B,E) and low level of hsa-miR-429 (*p* < 0.001 and *p* < 0.001) (Figure 6C,F) in both UBUC and UTUC (Table 1 and Table 2). However, high microvascular density was only evidently related to high mitotic rate in UTUC (*p* = 0.005) while being marginally correlated in UBUC (*p* = 0.138) (Table 1 and Table 2). Furthermore, the univariate analysis indicated that disease-specific survival (DSS) and metastasis-free survival (MeFS) were negatively determined by aggressive pT status (T), the presence of nodal metastasis, higher histological grade, the presence of vascular invasion and perineural invasion, higher expression of CEBPD and VEGFA (Figure 7A,B,E,F) and higher microvascular density (Figure 7C,D,G,H) in UBUC and UTUC. Yet, the lower expression of hsa-miR-429 confers a better prognosis. Moreover, DSS and MeFS represented a relation to high mitotic rate only in UBUC instead of UTUC patients under univariate analysis (Table 3 and Table 4). In multivariate analysis, UBUCs with high pT status (*p* = 0.003, R.R. = 12.308), high mitotic rate (*p* = 0.026, R.R. = 2.173) and high microvascular density (*p* = 0.021, R.R. = 3.509) showed significant worse DSS. The higher mitotic count (*p* = 0.036, R.R. = 1.721) and higher microvascular density (*p* < 0.001, R.R. = 11.494) were the most significant determinants for worse MeFS (Table 3). For UTUC, those with the presence of multifocality (*p* = 0.001, R.R. = 2.748), nodal metastasis (*p* < 0.001, R.R. = 3.802), perineural invasion (*p* = 0.009, R.R. = 2.799) and microvascular density (*p* = 0.003, R.R. = 4.596) had deteriorated DSS. The multivariate survival analysis for MeFS disclosed numerous factors including multifocality (*p* = 0.002, R.R. = 2.488), higher primary tumor status (*p* = 0.050, R.R. = 3.731), vascular invasion (*p* = 0.010, R.R. = 2.346), higher CEBPD (*p* < 0.001, R.R. = 3.886) and higher microvascular density (*p* = 0.001, R.R. = 5.611), which are significant independent survival determinants. The above data disclosed that the high status of CEBPD, VEGFA and microvascular density along with a low level of hsa-miR-429 strongly associated with aggressiveness and adverse survival rate.

## 4. Discussion

The novel correlation of the CEBPD–hsa-miR-429–VEGFA axis on UC angiogenesis and its important clinical significance were disclosed unprecedentedly in our present study. We discovered that CEBPD failed to increase the expression of VEGFA and capacity of HUVECs tube formation after the treatment of hsa-miR-429 mimic in BFTC909 and TCCSUP cell lines. However, hsa-miR-429 inhibitor could offset the depression of hsa-miR-429 mimic on mRNA expression of VEGFA and tube formation in CEBPD-overexpressing cells. Furthermore, UCs with the substantial expression of VEGFA and high microvascular density tended to be more aggressive and pursued worse clinical outcomes on account of the possession of higher expression of CEBPD and lower hsa-miR-429.

CEBPD is a well-known transcription factor and holds dual traits as a tumor suppressor in hepatocellular carcinoma [18], prostate cancer cells [19] or oncogene in lung adenocarcinoma [20], breast cancer [21] and UC [15] based on the diverse microenvironments of tumors [14,22,23,24,25]. In our previous study, we also fortified the oncogenic characteristic of CEBPD on UC specimens [10]. Owning to the multifaceted CEBPD of neoplasm, it is obligatory to excavate its widespread mechanism. To date, the reduction of MKI67 (cell proliferation marker) and PECAM1 (endothelial cell marker) has been observed in the *Cebpd*-deficient mice. Conditioned media incubated with *CEBPD*-knockdown cells also diminished the HUVECs tube formation [25], implying the vasculature development could be stimulated through the CEBPD expression.

miRNAs are pivotal for normal physiologic processes [26], while aberrant miRNAs are characterized as vital elements for tumor promotion or suppression depending on the different types of cancers [27]. hsa-miR-429 is a member of the hsa-miR-200 family. The loss of hsa-miR-429 mainly existed in the UBUCs with high grade and was strongly linked to adverse clinicopathological features and worse outcomes [28]. Consistently, a high level of hsa-miR-429 was detected in the low-grade UBUC cell line that inhibits cell invasion through the downregulation of proteins related to epithelial–mesenchymal transition (EMT) that enables tumor progression and metastasis [29]. Moreover, hsa-miR-429 was also depicted to act as an antimetastatic miRNA through the depletion of Raf/MEK/ERK-EMT pathway in hepatocellular carcinoma [30].

Administration of gemcitabine plus cisplatin is now considered as a first-line chemotherapy regimen for advanced or metastatic UC [31], but the prognostic outcome is still unsatisfactory, and the demand for combined remedy is pressing. Metastasis progression and nourishment supply strongly rely on vascularization in tumors [32]. VEGFA, a member of VEGF, binds to tyrosine kinase receptors (VEGFR-1 and VEGFR-2) and is a leading contributor to the formation of new blood vessels and promotes vascular permeability [33,34]. In cancer, VEGFA mainly comes from endothelial cells, hypoxic tumor cells and tumor-associated macrophages [35]. VEGFA-modulated neoangiogenesis has been connected to the EMT-triggered tumor initiation from cancer stemness [36]. The expression of VEGFA in bladder cancer specimens was substantially increased compared to those of normal mucosa (KOPPARAPU et al., 2013) and is regarded as a predictively prognostic biomarker in bladder cancer [37]. Patients who have higher *VEGFA* mRNA-expressing tumors usually showed an unfavorable survival [38]. Given this situation, the prominent role of VEGFA for tumor vasculature has been recognized as a feasible target for anticancer treatment. To date, various anti-VEGF–VEGFR agents such as anti-VEGFA neutralizing monoclonal antibodies (e.g., Bevacizumab, Aflibercept), anti-VEGFR antibodies (e.g., Ramucirumab), VEGFR tyrosine kinase inhibitors (e.g., Vandetanib) and small-molecule inhibitors of tyrosine kinase inhibitors (e.g., Sunitinib, Sorafenib, Pazopanib) have undergone clinical trials for solid tumor [39]. Combined first-/second-line chemotherapy with bevacizumab substantially raised the overall survival (OS) and progression-free survival (PFS) in patients with metastatic colorectal cancer (mCRC) and advanced cervical cancer [40,41] under a two-arm randomized phase III trial. However, adding ramucirumab to docetaxel in platinum-refractory UC represented favorable PFS yet had no significant effect on OS for RANGE trial [42]. Albeit a single-arm phase II trial for the combination of first-line chemotherapy (gemcitabine and cisplatin, GC) with bevacizumab significantly prolonged UC patients’ OS [43], the randomized phase III CALGB 90,601 (Alliance) study indicated that adding bevacizumab to GC treatment in UC patients had no strikingly effect on OS compared to those of the placebo group (GC treatment only) [44].

Our data suggested that the MVD, the consequence of tumoral angiogenesis, is significantly diverse in UC. Accordingly, a biomarker-based selection is mandatory to identify patients, which might be most beneficial via antiangiogenic therapeutics. Noticeably, antiangiogenic therapy has been reported to provoke hypoxia and inhibit nutrient supply. Consequently, neoplasm diverted its microenvironment toward hyperglycolysis to build a metabolic symbiosis: hypoxic tumor cells enforce the capacity of glycolysis, promoting the production of lactate and largely exported lactate via monocarboxylate transporter 4 (MCT4). Furthermore, tumor cells in the oxygen-sufficient regions utilize extracellular lactate to fuel the citric acid cycle for the accelerated oxidative phosphorylation. The metabolic symbiosis of hypoxic and normoxic tumor cells leads to the therapeutic resistance to antiangiogenic therapy. However, metabolic symbiosis collapses once the suppression of hyperglycolysis and lactate export [45]. Interestingly, our studies disclosed the CEBPD–has-miR429 signaling axis not only presented a proangiogenic property by regulating VEGFA but also promoted glycolysis through MYC-dependent and -independent signaling pathways. Moreover, we also demonstrated the expression of CEBPD could be remarkably upregulated by cisplatin in UC [15], which might partly explain the lack of survival advantage by the combination of gemcitabine, cisplatin and bevacizumab in the CALGB 90,601 (Alliance) trial. Taken together, a series of our studies suggested a more precise drug combination targeting angiogenesis could avoid activation of signaling cascades carrying resistance. Moreover, a biomarker-based design of antiangiogenic therapy and searching ways for CEBPD blockade for UC would be warranted.

Although there are current advantages of screening and remedy, the therapy still confronts many obstacles to overcome on account of the high heterogeneity of UC with the unfavorable outcome. In this study, we figured out the regulatory mechanism behind the VEGFA-induced neoangiogenesis through CEBPD-dependent hsa-miR-429 inhibition in UC. Accordingly, combining CEBPD with VEGFA as predictive strategies and therapy markers could tailor and enlarge the preventive and clinically prognostic management. This study could be beneficial to target the patients likely to apply for anti-VEGF targeted therapy in the future.

## Figures and Tables

**Figure 1 cells-11-00638-f001:**
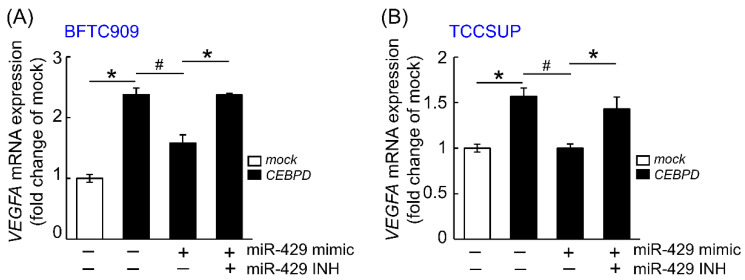
The inhibition effect of hsa-miR-429 mimic on *VEGFA* transcript is rescued after the combined treatment of hsa-miR-429 inhibitor in CEBPD-overexpressing BFTC909 and TCCSUP. Quantitative RT-PCR indicates that the mRNA level of *VEGFA* is evidently increased in the BFTC909 (**A**) and TCCSUP (**B**) cells with ectopic CEBPD expression than mock delivery. The transcript of *VEGFA* is significantly inhibited after the treatment of 50 nM hsa-miR-429 mimic for 72 h compared to that incubated of miRNA Mimic Negative Control in CEBPD-overexpressing cells. The combined transfection of hsa-miR-429 mimic (50 nM) and hsa-miR-429 inhibitor (50 nM) for 72 h recover the impact of CEBPD on the increase in the mRNA level of *VEGFA* in BFTC909 and TCCSUP cells. Data were executed in triplicate, and the statistical graphics are shown as the mean ± SEM. Statistical significance: * ^#^
*p* < 0.05. * is for the significant upregulation while ^#^ is for the significant downregulation.

**Figure 2 cells-11-00638-f002:**
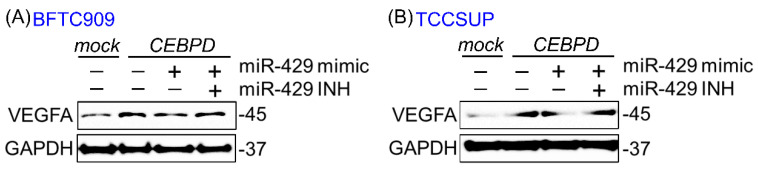
Downregulation of VEGFA protein expression by hsa-miR-429 mimic is rescued by the hsa-miR-429 inhibitor in the CEBPD-overexpressing BFTC909 and TCCSUP. Immunoblotting shows that the lentiviral-delivered CEBPD overexpression robustly promotes the protein level of VEGFA in BFTC909 (**A**) and TCCSUP (**B**) cells than in the mock-expressing cell lines. Upregulated VEGFA protein induced by the CEBPD is diminished after the transfection of 50 nM hsa-miR-429 mimic for 72 h in these two cells. A quantity of 50 nM hsa-miR-429 inhibitor could offset the depressing effect of 50 nM hsa-miR-429 mimic on the VEGFA protein in CEBPD-infected cells. GAPDH is served as a loading control.

**Figure 3 cells-11-00638-f003:**
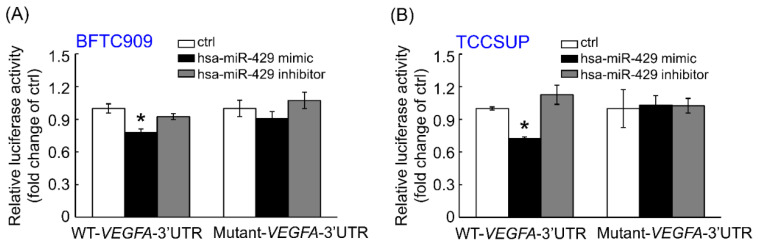
hsa-miR-429 regulates the expression of *VEGFA* transcript through targeting its 3′-UTR. The relative luciferase activity of wild type-*VEGFA*-3′-UTR reporter is strikingly decreased after the transfection of 50 nM hsa-miR-429 mimic for 72 h in both BFTC909 (**A**) and TCCSUP (**B**) cells compared to those of the control group. Nevertheless, the hsa-miR-429 mimic has no significant effect on the wild type-*VEGFA*-3′-UTR-Luc upon the presence of the 50 nM hsa-miR-429 inhibitor in these two cells. Furthermore, neither hsa-miR-429 mimic nor the combination of hsa-miR-429 mimic and inhibitor change the relative luciferase activity of mutant-*VEGFA*-3′-UTR in BFTC909 and TCCSUP. All data were conducted in triplicate and are represented as the mean ± SEM. Statistical significance: * *p* < 0.05.

**Figure 4 cells-11-00638-f004:**
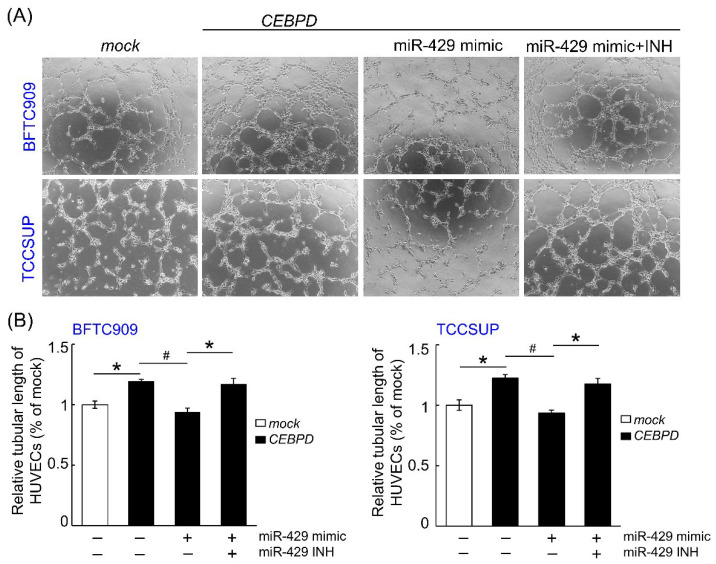
CEBPD overexpression-induced HUVEC tube forming in BFTC909 and TCCSUP is restrained by hsa-miR-429 mimic and is rescued by has-miR-429 inhibitor. (**A**) The ability of tube formation is markedly increased when the HUVECs are incubated with the conditioned medium from the CEBPD-overexpressing BFTC909 and TCCSUP than from the mock-groups. Nevertheless, conditioned medium from the CEBPD-overexpressing cells after the transfection of 50 nM hsa-miR-429 mimic for three days destroys the capacity of HUVECs tube formation. However, the conditioned medium collected from the BFTC909 and TCCSUP transfected with combined 50 nM hsa-miR-429 mimic and inhibitor for 3 days resists the repressed effect of the hsa-miR-429 mimic transfection alone on HUVECs tube forming. The statistical data are shown in (**B**). All experimental data were proceeded in triplicate and are shown as the mean ± SEM. Statistical significance: * ^#^
*p* < 0.05. * is for the significant upregulation while ^#^ is for the significant downregulation.

**Figure 5 cells-11-00638-f005:**
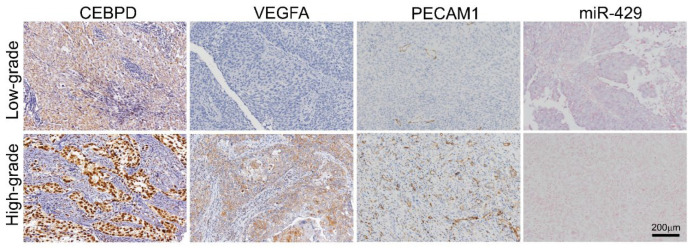
Higher expression of CEBPD is associated with lower hsa-miR-429, higher VEGFA expression, higher PECAM1-labeled microvascular density (MVD) and higher histological grade in UC. Immunohistochemistry shows that a higher level of CEBPD is associated with lower hsa-miR-429 expression. Of note, CEBPD expression is also correlated with high VEGFA expression, high PECAM1-labeled MVD and high histological grade in UC.

**Figure 6 cells-11-00638-f006:**
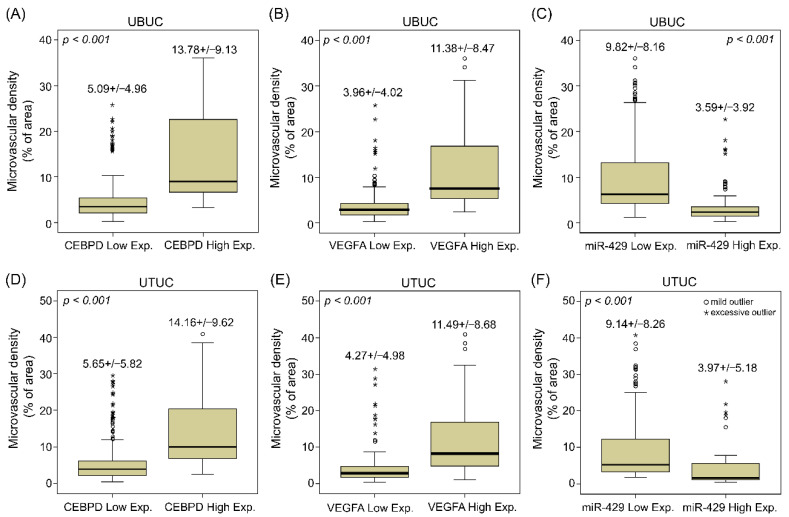
The expression of CEBPD and VEGFA have a positive correlation, but hsa-miR-429 represents a negative relation with MVD in UBUC and UTUC. Microvascular density is quantified by computerizing PECAM1-labeled vessels from IHC staining in UBUC and UTUC specimens using ImageJ software, which evaluated the proportion of vascular field in each representative area. Quantitative results show the UBUC and UTUC samples with higher CEBPD (**A**,**D**) and VEGFA (**B**,**E**), but lower levels of hsa-miR-429 (**C**,**F**) tend to have higher MVD. A circle symbol indicates a mild outlier (1.5 box lengths from the hinge of the box), and an asterisk is used to point an excessive outlier (3 box lengths from the hinge of the box).

**Figure 7 cells-11-00638-f007:**
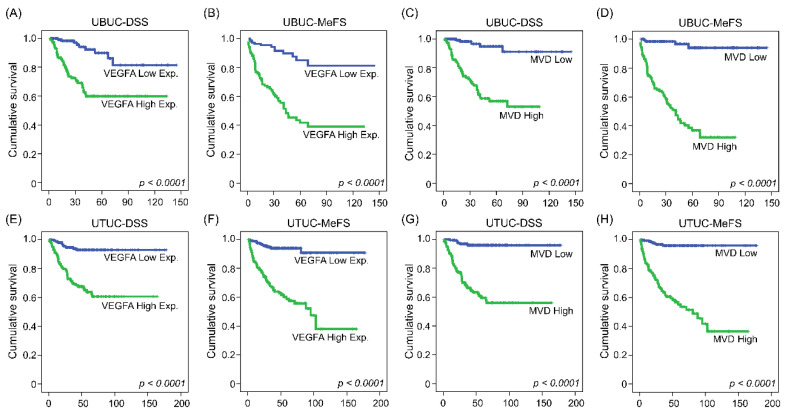
High expression of VEGFA and MVD are correlated with inferior survival outcomes in UBUC and UTUC patients. Kaplan–Meier plot examined that a higher level of VEGFA or MVD confers deteriorated clinical outcomes in terms of disease-specific survival and metastatic-free survival in patients with UBUC (**A**–**D**) and UTUC (**E**–**H**).

**Table 1 cells-11-00638-t001:** Correlations between CEBPD, miR-429 and VEGFA expression, microvascular density and other important clinicopathological parameters in UBUC.

Parameter	Category	Case No.	VEGFA Expression			Microvascular Density		
			Low	High	*p* Value	Low	High	*p* Value
Sex ^&^	Male	216	111	105	0.376	108	108	0.923
	Female	79	36	43		39	40	
Age (years) ^&^	<65	121	60	61	0.944	61	60	0.867
	≥65	174	87	87		86	88	
Primary tumor (T) ^&^	Ta	84	70	14	<0.001 *	73	11	<0.001 *
	T1	88	51	37		51	37	
	T2–T4	123	26	97		23	100	
Nodal metastasis ^&^	Negative (N0)	266	143	123	<0.001 *	147	119	<0.001 *
	Positive (N1–N2)	29	4	25		0	29	
Histological grade ^&^	Low grade	56	46	10	<0.001 *	43	13	<0.001 *
	High grade	239	101	138		104	135	
Vascular invasion ^&^	Absent	246	140	106	<0.001 *	140	106	<0.001 *
	Present	49	7	42		7	42	
Perineural invasion ^&^	Absent	275	145	130	<0.001 *	145	130	<0.001 *
	Present	20	2	18		2	18	
Mitotic rate (per 10 high power fields) ^#^		295	11.3 ± 14.12	17.5 ± 13.30	<0.001 *	13.2 ± 15.64	15.6 ± 12.18	0.138
CEBPD expression ^&^	Low expression	207	140	67	<0.001 *	144	63	<0.001 *
	High expression	88	7	81	-	3	85	
miR-429 expression ^&^	High expression	194	64	130	<0.001 *	66	128	<0.001 *
	Low expression	101	83	18		81	20	
VEGFA expression ^&^	Low expression	147	-	-	-	120	27	<0.001 *
	High expression	148	-	-	-	27	121	

^&^ Chi-square test; ^#^ Mann–Whitney U test; all other comparisons; * Statistical significance. Tumor samples were taken from the biobank of Chi Mei Medical Center.

**Table 2 cells-11-00638-t002:** Correlations between CEBPD, miR-429 and VEGFA expression, microvascular density and other important clinicopathological parameters in UTUC.

Parameter	Category	Case No.	VEGFA Expression			Microvascular Density		
			Low	High	*p* Value	Low	High	*p* Value
Sex ^&^	Male	158	76	82	0.514	76	82	0.514
	Female	182	94	88		94	88	
Age (years) ^&^	<65	138	80	58	0.015 *	77	61	0.077
	≥65	202	90	112		93	109	
Multifocality ^&^	Single	278	145	133	0.092	141	137	0.574
	Multifocal	62	25	37		29	33	
Primary tumor (T) ^&^	Ta	89	68	21	<0.001 *	70	19	<0.001 *
	T1	92	61	31		63	29	
	T2–T4	159	41	118		37	122	
Nodal metastasis ^&^	Negative (N0)	312	168	144	<0.001 *	168	144	<0.001 *
	Positive (N1–N2)	28	2	26		2	26	
Histological grade ^&^	Low grade	56	42	14	0.015 *	40	16	<0.001 *
	High grade	284	128	156		130	154	
Vascular invasion ^&^	Absent	234	150	84	<0.001 *	157	77	0.001 *
	Present	106	20	86		13	93	
Perineural invasion ^&^	Absent	321	169	152	<0.001 *	169	152	<0.001 *
	Present	19	1	18		1	18	
Mitotic rate (per 10 high power fields) ^#^		340	10.8 ± 12.02	13.8 ± 12.38	0.022 *	10.5 ± 11.58	14.16 ± 12.71	0.005 *
CEBPD expression ^&^	Low expression	251	158	93	<0.001 *	153	98	<0.001 *
	High expression	89	12	77		17	72	
miR-429 expression ^&^	High expression	257	109	148	<0.001 *	113	144	<0.001 *
	Low expression	83	61	22	-	57	26	
VEGFA expression ^&^	Low expression	170	-	-	-	129	41	<0.001 *
	High expression	170	-	-		41	129	

^&^ Chi-square test; ^#^ Mann–Whitney U test; * Statistical significance. Tumor samples were taken from the biobank of Chi Mei Medical Center.

**Table 3 cells-11-00638-t003:** Univariate log-rank and multivariate analyses for disease-specific and metastasis-free survival in UBUC.

Parameter	Category	Case No.	Disease-Specific Survival					Metastasis-Free Survival				
			Univariate Analysis		Multivariate Analysis			Univariate Analysis		Multivariate Analysis		
			No. of Events	*p* Value	RR	95% CI	*p* Value	No. of Events	*p*-Value	RR	95% CI	*p* Value
Sex	Male	216	41	0.4906	-	-	-	61	0.2745	-	-	-
	Female	79	11		-	-	-	16		-	-	-
Age (years)	<65	121	17	0.1315	-	-	-	32	0.8786	-	-	-
	≥65	174	35		-	-	-	45		-	-	-
Primary tumor (T)	Ta	84	1	<0.0001 *	1	-	0.003 *	4	<0.0001 *	1	-	0.491
	T1	88	9		3.532	0.343–36.331		23		2.126	0.533–8.176	
	T2–T4	123	42		12.308	1.194–126.906		50		2.344	0.576–9.529	
Nodal metastasis	Negative (N0)	266	41	0.0001 *	1	-	0.923	61	<0.0001 *	1	-	0.123
	Positive (N1–N2)	29	11		1.036	0.507–2.115		16		1.614	0.878–2.966	
Histological grade	Low grade	56	2	0.0016 *	1	-	0.809	5	0.0007 *	1	-	0.576
	High grade	239	50		0.814	0.153–4.326		72		1.363	0.460–4.039	
Vascular invasion	Absent	246	37	0.0010 *	1	-	0.108	54	<0.0001 *	1	-	0.726
	Present	49	15		0.558	0.274–1.137		23		0.898	0.492–1.640	
Perineural invasion	Absent	275	44	<0.0001 *	1	-	0.065	67	0.0003 *	1	-	0.311
	Present	20	8		2.252	0.952–5.327		10		1.491	0.689–3.227	
Mitotic rate(per 10 high power fields)	<10	139	12	0.0001 *	1	-	0.026 *	23	<0.0002 *	1	-	0.036 *
	≥10	156	40		2.173	1.097–4.304	-	54		1.721	1.035–2.864	-
CEBPD expression	Low	207	22	<0.0001 *	1	-	0.259	32	<0.0001 *	1	-	0.452
	High	88	30		1.446	0.762–2.742		45		1.216	0.730–2.026	
miR-429 expression	Low	194	48	<0.0001*	1	-	0.273	66	<0.0001 *	1	-	0.987
	High	101	4		0.542	0.182–1.618		11		1.006	0.499–2.026	
VEGFA expression ^&^	Low expression	147	9	<0.0001 *	-	-	0.980	13	<0.0001 *	-	-	0.258
	High expression	148	43		1.012	0.414–2.471	-	64		1.501	0.742–3.035	-
Microvascular density	Low	147	5	<0.0001 *	1	-	0.021 *	4	<0.0001 *	1	-	<0.001
	High	148	47		3.509	1.208–10.196		73		11.494	3.825–34.542	

^&^ Chi-square test; * Statistical significance.

**Table 4 cells-11-00638-t004:** Univariate log-rank and multivariate analyses for disease-specific and metastasis-free survival in UTUC.

Parameter	Category	Case No.	Disease-Specific Survival					Metastasis-Free Survival				
			Univariate Analysis		Multivariate Analysis			Univariate Analysis		Multivariate Analysis		
			No. of Events	*p* Value	RR	95% CI	*p* Value	No. of Events	*p* Value	RR	95% CI	*p* Value
Sex	Male	158	28	0.9301	-	-	-	32	0.7904	-	-	-
	Female	182	33		-	-	-	38		-	-	-
Age (years)	<65	138	26	0.8660	-	-	-	30	0.8470	-	-	-
	≥65	202	35		-	-	-	40		-	-	-
Multifocality	Single	273	43	0.0042 *	1	-	0.001 *	52	0.0196 *	1	-	0.002 *
	Multifocal	62	18		2.748	1.516–4.984		18		2.488	1.175–3.665	
Primary tumor (T)	Ta	89	2	<0.0001 *	1	-	0.088	4	<0.0001 *	1	-	0.050 *
	T1	92	9		3.642	0.726–18.220		15		1.924	0.580–6.378	
	T2–T4	159	50		4.162	0.877–19.757		51		3.731	1.141–12.197	
Nodal metastasis	Negative ((N0)	312	42	<0.0001 *	1	-	<0.001 *	55	<0.0001 *	1	-	0.104
	Positive (N1–N2)	28	19		3.802	1.998–7.234		15		1.711	0.895–3.271	
Histological grade	Low grade	56	4	0.0171 *	1	-	0.105	3	0.0019 *	1	-	0.398
					2.580	0.820–8.122				1.733	0.484–6.206	
Vascular invasion	Absent	234	24	<0.0001 *	1	-	0.458	26	<0.0001 *	1	-	0.010 *
	Present	106	37		1.268	0.677–2.376		44		2.346	1.231–4.469	
Perineural invasion	Absent	321	50	<0.0001 *	1	-	0.009 *	61	<0.0001 *	1	-	0.238
	Present	19	11		2.799	1.296–6.048		9		1.599	0.733–3.487	
Mitotic rate(per 10 high power fields)	<10	173	27	0.1268	-	-	-	30	0.0581	-	-	-
	≥10	167	34		-	-	-	40		-	-	-
CEBPD expression	Low	251	26	<0.0001 *	1	-	0.081	24	<0.0001 *	1	-	<0.001 *
	High	89	35		1.740	0.934–3.243		46		3.886	2.119–7.125	
miR-429 expression	Low	257	55	0.0024 *	1	-	0.908	64	<0.0001 *	1	-	0.633
	High	83	6		1.055	0.423–2.633		6		1.250	0.500–3.127	
VEGFA expression ^&^	Low expression	170	10	<0.0001 *	-	-	0.743	10	<0.0001 *	-	-	0.660
	High expression	170	51		0.868	0.371–2.029		60		1.202	.530–2.724	
Microvascular density	Low	170	6	<0.0001 *	1	-	0.003 *	6	<0.0001 *	1		0.001 *
	High	170	55		4.596	1.666–12.679		64		5.611	2.102–14.976	

^&^ Chi-square test; * Statistical significance.

## Data Availability

Not applicable.

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
