# Peer review of "Angiogenesis Driven by the CEBPD–hsa-miR-429–VEGFA Signaling Axis Promotes Urothelial Carcinoma Progression"

_cells, 2022, doi:10.3390/cells11040638_

Round 1
Reviewer 1 Report
What is the mechanism of the hsa-miR-429 inhibitor? Is the inhibitor really specific to hsa-miR-429?
Fig 1 showed that miR-429 inhibit VEGFA expression induced by CEBPD. However, why is VEGFA positively associated with miR-429 expression in table.1?
Authors described that high status of CEBPD, VEGFA and microvascular density along with a low level of 311 hsa-miR-429 strongly associated with aggressiveness and adverse survival rate. However, given that the multivariate analysis revealed no significant correlation of VEGFA and has-miR-429 with prognosis, it seems like that these factors only indirectly, but not strongly, associated with adverse survival rate.
Author Response
Dear Reviewer,
Please see the attachment.
Many thanks,
Chien-Feng

Reviewer 2 Report
The authors present their work on deciphering the modulation of the CEBPD/hsa-miR-429/VEGFA axis on the progression of UC. This work is potentially interesting, however the clinical significance is unclear,
1. Particularly given the negative results of CALGB study (Randomized Phase III Trial of Gemcitabine and Cisplatin With Bevacizumab or Placebo in Patients With Advanced Urothelial Carcinoma: Results of CALGB 90601 (Alliance) | Journal of Clinical Oncology (ascopubs.org). This should be discussed.
2. Also VEGF WB panel in figure 2 is blurry and needs to be replaced.
Author Response

(The authors gave the same response as above.)

Round 2
Reviewer 1 Report
I have no more comments.